# MATHCODER2: BETTER MATH REASONING FROM CONTINUED PRETRAINING ON MODEL-TRANSLATED MATHEMATICAL CODE

**Zimu Lu**[*1], **Aojun Zhou**[*1], **Houxing Ren**[1], **Ke Wang**[1], **Weikang Shi**[1]
**Junting Pan**[1,2], **Mingjie Zhan**[†1], **Hongsheng Li**[†1,2]
[1]Multimedia Laboratory (MMLab), The Chinese University of Hong Kong     [2]CPII under InnoHK
`luzimu@mail.ustc.edu.cn`  `{aojunzhou, zmjdll}@gmail.com`
`hsli@ee.cuhk.edu.hk`

## ABSTRACT

Code has been shown to be effective in enhancing the mathematical reasoning abilities of large language models due to its precision and accuracy. Previous works involving continued mathematical pretraining often include code that utilizes math-related packages, which are primarily designed for fields such as engineering, machine learning, signal processing, or module testing, rather than being directly focused on mathematical reasoning. In this paper, we introduce a novel method for generating mathematical code accompanied with corresponding reasoning steps for continued pretraining. Our approach begins with the construction of a high-quality mathematical continued pretraining dataset by incorporating math-related web data, code using mathematical packages, math textbooks, and synthetic data. Next, we construct reasoning steps by extracting LaTeX expressions, the conditions needed for the expressions, and the results of the expressions from the previously collected dataset. Based on this extracted information, we generate corresponding code to accurately capture the mathematical reasoning process. Appending the generated code to each reasoning step results in data consisting of paired natural language reasoning steps and their corresponding code. Combining this data with the original dataset results in a 19.2B-token high-performing mathematical pretraining corpus, which we name MathCode-Pile. Training several popular base models with this corpus significantly improves their mathematical abilities, leading to the creation of the MathCoder2 family of models. All of our data processing and training code is open-sourced, ensuring full transparency and easy reproducibility of the entire data collection and training pipeline.

## 1 INTRODUCTION

Various studies (Azerbayev et al., 2024; Shao et al., 2024) have shown that training on code enhances the mathematical reasoning abilities of large language models (LLMs). Previous research in continued mathematical pretraining often includes code that utilizes math-related packages (Azerbayev et al., 2024). This code is typically sourced from GitHub and is primarily designed for fields such as engineering, machine learning, signal processing, or module testing, rather than focusing directly on mathematics. Recent models (Zhou et al., 2024; Yang et al., 2024b; Ying et al., 2024; Shao et al., 2024; Wang et al., 2023a) have adopted Tool-Integrated Reasoning (TIR) in fine-tuning. They utilize integrated natural language reasoning steps and Python code to improve performance on mathematical reasoning tasks. Reasoning with the help of code is particularly effective for more challenging problems, likely due to its precision and accuracy.

Although utilizing existing open-source code in the pretraining phase can enhance the mathematical reasoning abilities of LLMs, such code often lacks accompanying natural language explanations or context. This might hinder the model's ability to fully understand them. In this paper, we propose a novel method for *generating large amounts of mathematical code accompanied by corresponding natural language reasoning steps*, which are extracted from math-related pretraining texts. Different

from the existing math-related code, our generated code is paired with natural language reasoning steps, making the code more comprehensible. Also, as our code is generated based on math-related texts, they are all highly related to mathematical reasoning. When used in pretraining, the mathematical code paired with reasoning steps facilitates LLMs' understanding of math-related pretraining texts, as it effectively captures the underlying reasoning process. Furthermore, this data enhances the model's potential to be finetuned for TIR reasoning.

Our data processing pipeline consists of two key steps: (1) carefully curating a robust basic dataset for pretraining, and (2) generating paired reasoning steps and mathematical code by extracting La-TeX expressions and their context, translating the extracted information into Python code snippets, executing the generated code snippets, and verifying their correctness.

First, we gather and carefully filter a wide variety of math-related data sources, including web pages, model-generated data, math-related code, and textbooks. Through an advanced filtering process, we ensure the dataset is both large and highly relevant, minimizing irrelevant content while preserving the mathematical texts necessary for training. This results in a 16.5B-token dataset that forms the foundation of our pretraining efforts. By conducting experiments with smaller models, we show that this careful curation leads to more efficient training without sacrificing model performance.

Second, we propose a novel method for generating large amounts of paired mathematical reasoning steps and their corresponding Python code. Given a piece of text from the pretraining corpus collected above, we wrap it in a carefully designed prompt that instructs a Llama-3.1-70B-Instruct model to extract LaTeX expressions along with their relevant context, including the conditions for each expression and the result of its computation. This results in a list of comprehensive mathematical reasoning steps, complete with the necessary conditions, the computations taken, and the results. Then, we prompt the model to translate each reasoning step into a Python code snippet that captures the underlying reasoning process. The generated Python snippets are executed, and only those that run successfully and produce outputs matching the expected results are retained. By pairing the code with the corresponding reasoning step, we create the final data. The process is demonstrated in the lower half of Fig. 1. This process yields a 2.7B-token corpus of mathematical code snippets accompanied with their corresponding reasoning steps, which we combine with the data generated in the first step to create a 19.2B-token pretraining dataset, named **MathCode-Pile**.

We validate the effectiveness of MathCode-Pile on four popular base models: Llama-3-8B, DeepSeekMath-7B, Mistral-7B, and Code-Llama-7B, significantly improving their performance on five representative mathematical benchmarks. We name the resulting family of pretrained models MathCoder2. In particular, MathCoder2-Llama-3-8B achieves 4-shot accuracies of 38.4% on MATH and 69.9% on GSM8K, outperforming the baseline of training only on the basic data generated in the first step by 3.1% and 4.1%, respectively. This demonstrates that the data of mathematical code accompanied with reasoning steps effectively enhances LLMs' reasoning abilities.

Different from recent works, such as DeepSeekMath (Shao et al., 2024), InternLM-Math (Ying et al., 2024), and Qwen2.5-Math (Yang et al., 2024b), which only release their model weights, we offer a detailed, open-source framework for data processing and training that achieves performance competitive with these models, fostering further progress in mathematical reasoning for LLMs.

Our contributions include:

- A novel and effective method for generating large amounts of mathematical code with corresponding natural language reasoning steps, significantly enhancing pretraining outcomes.

- The creation of MathCode-Pile, a meticulously curated 19.2B-token dataset for continued mathematical pretraining. This dataset includes math-related web data, synthetic data, code, textbooks, and model-translated mathematical code.

- Full open-sourcing of all data processing and training code, ensuring transparency and reproducibility to support future research.

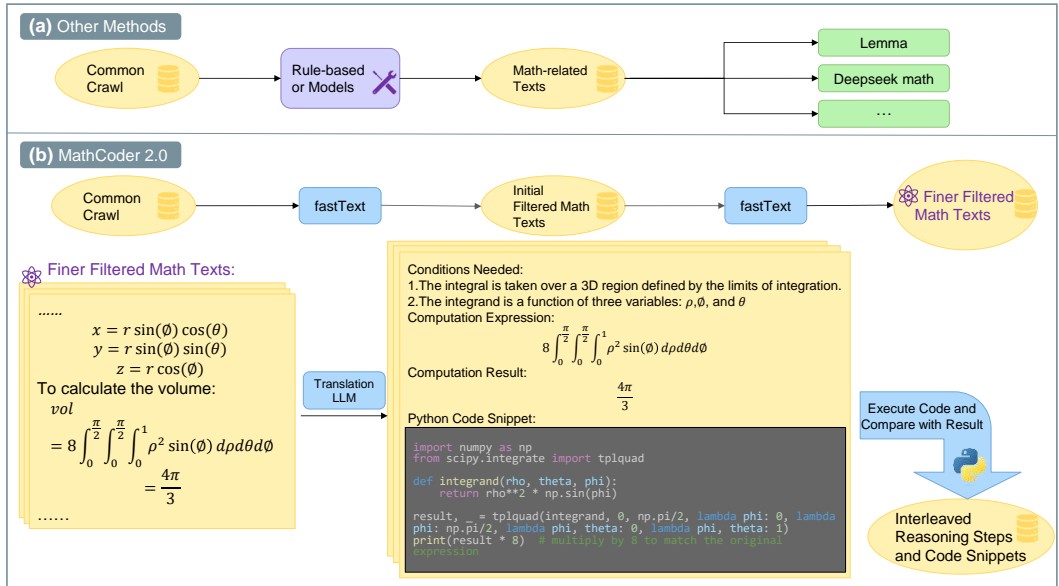

Figure 1: The data processing pipeline. (a) shows the pipeline of prior works. (b) demonstrates our method. We first use a fastText classifier to filter the Common Crawl corpus, resulting in the initial filtered math texts. Then, we annotate part of the filtered texts to train a new fastText classifier, and conduct a second filtering, resulting in the finer filtered math texts. Then we use an instruction-tuned LLM to extract reasoning steps from these math-related texts, and translate the reasoning steps into corresponding code snippets. We execute the code snippets and compare the output with the expected result. If the code executes successfully and the result is as expected, the code is retained.

# 2    CURATION OF MATHCODE-PILE

We curate our mathematical pretraining dataset, MathCode-Pile, in two steps: first, we collect the basic data in Sec. 2.1, and then we use it to generate mathematical code snippets with their corresponding natural language reasoning steps in Sec. 2.2.

## 2.1    BASIC DATA

We collect and carefully filter a diverse range of mathematical data to ensure relevance and quality for continued pretraining of LLMs. The data includes math-related web content, synthetic data, code utilizing mathematical packages, and mathematical textbooks.

**Math-related Web Data**.  Web data offers a broad range of real-world mathematical examples. We start with the OpenWebMath (Paster et al., 2023) dataset, which contains mathematical web pages sourced from Common Crawl. Observing that a significant portion of these documents are unrelated to mathematics, we instruct the Mixtral-8x7B-Instruct model with a carefully designed prompt (detailed in Appendix A) to filter out irrelevant texts. Examples of irrelevant texts are shown in Appendix D. This reduces the dataset from 13.7B tokens to 4.8B tokens (measured using the Llama-3 tokenizer). We call this filtered version filtered-OpenWebMath.

To further expand the dataset, we train a fastText classifier (Joulin et al., 2016) using filtered-OpenWebMath as positive samples and random Common Crawl data as negative samples (training details are explained in Appendix. B). This model helps identify additional math-related documents within the Common Crawl data from Matrix (Zhang et al., 2024), a general pretraining dataset. A second round of filtering is performed, where Mixtral-8x7B-Instruct annotates a portion of these documents, and a new fastText classifier trained based on these annotations further refines the data. This produces 6.4B tokens, which we label as filtered-CC-En-math. Finally, we combine filtered-OpenWebMath and filtered-CC-En-math, resulting in a comprehensive 11.2B-token math-related web dataset.

**Prompt:** You will be presented with a text related to math. I need you to identify all the complex computations in it. For each complex computation, find out the conditions needed for the computation, the LaTeX expression that conducts the computation, and the result of the computation. Then generate a Python code snippet for each computation that demonstrates how the result is reached. Output each computation in the following format:

Conditions Needed:
1. [Condition 1]
2. [Condition 2]
...

Computation Expression:
$[LaTeX Expression]$

Computation Result:
[Computation Result]

Python Code Snippet:

```python
[Python Code]
```

There can be more than one complex computation in the text. Output only the computations that requires calculation. Do not include mathematical statements or definitions as a computation. Make sure each snippet can be executed individually. The text is as follows: {TEXT}
The computations are:

Figure 2: The prompt for extracting reasoning steps from texts in the pretraining corpus and generating the corresponding Python snippets. {TEXT} is replaced with the text from the dataset collected in Sec. 2.1.

**Synthetic Data**. Synthetic data offers structured mathematical texts that complement the web data. We collect synthetic data from various open-source repositories on Hugging Face, including datasets like Education-College-Students[1], Maths-College[2], and synthetic math books from Matrix (Zhang et al., 2024). To ensure relevance, we apply a fastText classifier to filter out non-mathematical documents, refining the dataset to 2.2B tokens of high-quality synthetic math content.

**Code Utilizing Mathematical Packages**. Code data offers practical examples of how mathematical libraries are used in programming. We collect code from Python and Jupyter files within the Star-CoderData dataset (Li et al., 2023), retaining only programs that import math-related packages such as sympy, fractions, cmath, scipy, or statistics. The widely used numpy package is not used to filter the documents, as it appears frequently in non-mathematical contexts. After filtering, this collection process results in 1.7B tokens of code related to mathematical computations.

**Mathematical Textbooks**. Textbooks provide formal, structured presentations of mathematical concepts, making them a valuable source of math knowledge. We gather 8K PDFs of textbooks from online resources by identifying those with titles containing math-related keywords such as algebra, geometry, probability, etc. These PDF files are then converted into markdown format using the Nougat tool for easier integration into our training pipeline.

## 2.2 MODEL-TRANSLATED MATHEMATICAL CODE

In this section, we propose a novel approach for extracting reasoning steps from the basic pretraining data and translating them into corresponding Python code snippets that capture the underlying reasoning processes. This extraction and translation process is performed using a strong instruction-tuned model, which is Llama-3.1-70B-Instruct in this paper.

---

[1] https://huggingface.co/datasets/ajibawa-2023/Education-College-Students
[2] https://huggingface.co/datasets/ajibawa-2023/Maths-College

Table 1: The components and data statistics of MathCode-Pile.

| Components | Size (MB) | Documents | Tokens | Average (Tokens) |
|---|---|---|---|---|
| Filtered-OpenWebMath | 16,999 | 2,824,705 | 4,826,902,621 | 1,709 |
| Filtered-CC-En-math | 23,465 | 7,597,718 | 6,341,745,645 | 835 |
| Synthetic data | 8,855 | 2,195,974 | 2,193,189,314 | 999 |
| Code using math packages | 6,120 | 513,059 | 1,703,226,005 | 3,320 |
| Mathematical textbooks | 4,431 | 8,373 | 1,390,268,773 | 166,042 |
| **Translated mathematical code** | 8,235 | 6,347,823 | 2,728,740,985 | 430 |
| Total | 68,105 | 19,487,652 | 19,184,073,343 | 984 |

Our method begins with taking a piece of text from the basic pretraining data and wrapping it in a carefully designed prompt, as shown in Fig. 2. This prompt instructs the model to identify *LaTeX expressions* denoting complex computations, along with the necessary context, including the *conditions required for the computation* and the *expected result*. By explicitly extracting the conditions of the LaTeX expression, we enhance the model's ability to comprehend the underlying mathematical reasoning behind the usage of the expression. The expected result of the computation can later serve as a basis for verifying the correctness of the generated code. A mathematical reasoning step is constructed by combining the conditions, expression and result. The prompt then directs the model to produce a *Python code snippet* that accurately reflects the underlying reasoning process behind the extracted reasoning step. The model is asked to present the conditions, LaTeX expression, result, and Python code snippet in a structured format, ensuring that each part can be easily extracted from the generated text. Examples of generated texts are shown in Appendix C.

After the Python code snippets are generated, they are executed, and outputs of the execution are compared with the expected results extracted from the generated text. Only the Python code snippets that execute without errors and produce correct outputs are retained. This filtering process ensures a higher quality of generated code, making the resulting dataset more reliable for mathematical pretraining compared to approaches that rely on unverified and general-purpose code.

Leveraging the Llama-3.1-70B-Instruct model, we initially generated 3.1B tokens of the data. After applying the filtering process, we obtain a total of 2.7B tokens of high-quality data of mathematical code snippets accompanied with their corresponding reasoning steps. This newly generated data significantly enriches our original pretraining corpus. By combining this data with the basic pretraining data, we create a comprehensive pretraining dataset totaling 19.2B tokens, which we refer to as **MathCode-Pile**. Detailed statistics of MathCode-Pile are presented in Tab. 1. This dataset is tailored specifically for enhancing the mathematical and coding abilities of LLMs. To avoid benchmark contamination, we follow Yang et al. (2024b) to filter out samples that have significant overlaps with any of the questions from benchmark datasets used in evaluation. We use exact match to remove the identical samples and further apply 13-gram deduplication (with a condition that the Jaccard similarity should be larger than 0.6) to remove more samples that might cause contamination.

In comparison to traditional methods of curating math-related code, which often draw on general-purpose repositories, our method ensures that the code is not only syntactically correct but also mathematically sound, reflecting a deeper understanding of mathematical reasoning. Our mathematical code is accompanied with corresponding natural language reasoning steps, which makes understanding the reasoning process easier. This makes MathCode-Pile a superior resource for models aimed at performing advanced mathematical reasoning tasks.

## 3 EXPERIMENTS

To demonstrate the effectiveness of our method, we first train several base models ranging from 7B to 8B parameters using MathCode-Pile and compare them to other best-performing models of the same size. The group of models resulting from the continued mathematical pretraining is named **MathCoder2**. Next, we train and compare various other open-source math pretraining datasets against MathCode-Pile using a smaller model, DeepSeekCoder-1.3B. To showcase the potential of

Table 2: Performance of various pretrained models on five representative mathematical datasets. All results reported are based on greedy decoding. "Code-open" shows whether the code for data-processing and model-training is open-sourced. The red numbers show the improvements compared to the base model from which each MathCoder2 model is trained.

| Model | Size | Code-open | MATH | GSM8K | SAT | OCW | MMLU-MATH |
|---|---|---|---|---|---|---|---|
| Qwen2-Math | 7B | ✗ | 50.4 | 80.4 | 87.5 | 14.0 | 57.9 |
| Qwen2.5-Math | 7B | ✗ | 55.4 | 91.6 | - | - | - |
| InternLM2.5 | 7B | ✗ | 34.0 | 74.8 | 65.6 | 8.1 | 49.6 |
| InternLM2-Math-Base | 7B | ✗ | 21.5 | 49.2 | - | - | - |
| Llemma | 7B | ✓ | 18.0 | 36.4 | 53.1 | 7.7 | - |
| Llama-2 | 7B | ✗ | 3.2 | 11.8 | 25.0 | 3.7 | - |
| Llama-3 | 8B | ✗ | 21.4 | 54.8 | 56.3 | 10.3 | 42.8 |
| **MathCoder2-Llama-3** | 8B | ✓ | 38.4(+17.0) | 69.9(+15.1) | 84.4(+28.1) | 18.0(+7.7) | 46.5(+3.7) |
| DeepSeekMath | 7B | ✗ | 36.2 | 64.2 | 84.4 | 15.4 | 47.4 |
| **MathCoder2-DeepSeekMath** | 7B | ✓ | 38.6(+2.4) | 68.8(+4.6) | 90.6(+6.2) | 16.9(+1.5) | 48.3(+0.9) |
| Mistral | 7B | ✗ | 13.1 | 52.2 | 75.0 | 8.5 | 38.3 |
| **MathCoder2-Mistral** | 7B | ✓ | 36.7(+23.6) | 68.2(+16.0) | 81.3(+6.3) | 13.2(+4.7) | 42.2(+3.9) |
| Code-Llama | 7B | ✗ | 6.7 | 14.6 | 25.0 | 3.7 | 26.4 |
| **MathCoder2-Code-Llama** | 7B | ✓ | 28.8(+22.1) | 52.3(+37.7) | 71.9(+46.9) | 8.5(+4.8) | 33.7(+7.3) |

the MathCoder2 models, we further perform supervised fine-tuning on them. Finally, we conduct ablation studies to analyze the impact of each component of the dataset.

## 3.1 MAIN RESULTS

**Benchmark datasets.** We evaluate the MathCoder2 models on five representative datasets: GSM8K (Cobbe et al., 2021), MATH (Hendrycks et al., 2021b), SAT-Math (Azerbayev et al., 2024), OCW (Lewkowycz et al., 2022), and MMLU-Math (Hendrycks et al., 2021a). GSM8K and MATH are tested using a 4-shot prompt with MAmmoTH's evaluation framework (Yue et al., 2023). SAT-Math and OCW are tested using a 4-shot prompt with DeepSeekMath's evaluation framework (Shao et al., 2024). MMLU-Math is tested using the lm-evaluation-harness's (Gao et al., 2024) default zero-shot setting for MMLU. These datasets cover a wide range of mathematical problems across various types and difficulty levels, from primary school math word problems to college-level challenges, providing a comprehensive evaluation of the models.

**Base models and training settings.** To demonstrate that our pretraining corpus is effective across different base models, we continue pretraining four base models with MathCode-Pile: Llama-3-8B (Dubey et al., 2024), DeepSeekMath-7B (Shao et al., 2024), Mistral-7B (Jiang et al., 2023), and Code-Llama-7B (Rozière et al., 2024). MathCoder2-Llama-3-8B is trained for 3 epochs with a global batch size of 4 million tokens and an 8192 token context length. MathCoder2-DeepSeekMath, MathCoder2-Mistral, and MathCoder2-CodeLlama are each trained for 3 epochs with a global batch size of 4 million tokens and a 4096 token context length.

**Baselines.** We compare our method with various other base models that possess strong mathematical abilities and are of similar sizes, including Qwen2-Math 7B (Yang et al., 2024a), Qwen2.5-Math 7B (Yang et al., 2024b), InternLM2-Math-Base 7B (Ying et al., 2024), InternLM2.5 7B (Cai et al., 2024), DeepSeekMath 7B (Shao et al., 2024), Llemma 7B (Azerbayev et al., 2024), Mistral 7B (Jiang et al., 2023), Llama2 7B (Touvron et al., 2023), Llama3 8B (Dubey et al., 2024) and Code-Llama 7B (Rozière et al., 2024).

**Results**: As demonstrated in Tab. 2, continued pretraining on MathCode-Pile consistently improves performance across all five benchmark datasets. MathCoder2 models rival the performance of top models like InternLM2-Math-Base, InternLM2.5, and DeepSeekMath. In particular, MathCoder2-DeepSeekMath demonstrates that our method continues to enhance the performance of DeepSeek-Math, a model that has already been extensively trained on large amounts of math-related data. How-

Table 3: Performance of various finetuned models on five representative mathematical datasets. All results reported are based on greedy decoding.

| Model | Size | MATH | GSM8K | OCW | Olympiad Bench | SVAMP |
|---|---|---|---|---|---|---|
| Qwen2-Math-Instruct | 7B | 75.1 | 89.9 | 34.6 | 38.2 | - |
| Qwen2.5-Math-Instruct | 7B | 83.6 | 95.2 | 37.1 | 41.6 | - |
| DeepSeekMath-Instruct-CoT | 7B | 46.8 | 82.9 | - | - | - |
| DeepSeekMath-Instruct-TIR | 7B | 57.4 | 83.7 | - | - | - |
| InternLM2-math-plus | 7B | 54.4 | 84.0 | 17.3 | 18.8 | - |
| NuminaMath-7B-CoT | 7B | 55.2 | 75.4 | 19.1 | 19.9 | - |
| NuminaMath-7B-TIR | 7B | 68.1 | 84.6 | - | - | - |
| ToRA-Code | 7B | 44.6 | 72.6 | - | - | 70.4 |
| MathCoder | 7B | 30.2 | 67.8 | - | - | 70.7 |
| MAmmoTH2-Plus | 8B | 42.8 | 84.1 | - | - | - |
| Llama-3.1-Instruct | 8B | 47.2 | 76.6 | 21.7 | 15.4 | - |
| **MathCoder2-Llama-3-Instruct-CoT** | 8B | 58.5 | 83.9 | 29.4 | 25.8 | 92.7 |
| **MathCoder2-Llama-3-Instruct-TIR** | 8B | 69.7 | 85.8 | 37.6 | 37.6 | 94.9 |
| **MathCoder2-DeepSeekMath-Instruct-CoT** | 7B | 55.2 | 80.3 | 30.9 | 23.0 | 92.1 |
| **MathCoder2-DeepSeekMath-Instruct-TIR** | 7B | 69.6 | 86.5 | 41.9 | 37.9 | 92.8 |

ever, there remains a performance gap between MathCoder2 and the Qwen2-Math and Qwen2.5-Math models. This gap might be attributed to their superior computational, manual, and financial resources, which enable the scaling of data size and the further improvement of data quality, reporting a mathemtaical dataset of 700B tokens (Yang et al., 2024b).

In contrast to models like Qwen2-Math, which only open-source their model weights, with much of their data processing and training details undisclosed, MathCoder2 is fully open-sourced, including all data processing pipelines and training code. This openness facilitates transparency, reproducibility, and further research, which is crucial for advancing the field. Compared to Llemma, which also open-sources its code, our method achieves better performance on the five datasets. Particularly, when trained on the same base model, Code-Llama, our method performs significantly better, which demonstrates the effectiveness of the MathCode-Pile pretraining corpus.

## 3.2 POST-TRAINING

To further demonstrate the potential of the MathCoder2 models in aligning to mathematical problem-solving tasks, we select the MathCoder2-Llama-3-8B model and MathCoder2-DeepSeekMath-7B for finetuning on mathematical problem-solution pairs. We first train the base model on general mathematical instructions following Yue et al. (2024) for two epochs. Subsequently, we finetune the model on NuminaMath-CoT[3], and NuminaMath-TIR[4] datasets for three epochs.

The results are shown in Tab. 3. MathCoder2-Instruct-TIR achieves high performance on all five datasets, reaching 69.7% on MATH and 86.5% on GSM8K, outperforming many of the best open-source models of similar size and demonstrating our method's potential to improve performance on downstream mathematical reasoning tasks. As this paper focuses on continued mathematical pretraining, the post-training serves only as a validation of the potential of our models. We conducted only simple supervised fine-tuning, without performing reinforcement learning or direct preference optimization, which could further improve performance on downstream tasks.

## 3.3 ABLATION STUDIES

In this session, we first analyze the impact of various components of the training data. Next, we compare MathCode-Pile to other open-source mathematical pretraining corpora.

**Analysis of the impact of the mathematical code**. We analyze the impact of the mathematical code on continued pretraining by comparing the results of adding and not adding the mathematical code. As shown in Tab. 4, the addition of the mathematical code in the pretraining corpus significantly

---

[3]https://huggingface.co/datasets/AI-MO/NuminaMath-CoT
[4]https://huggingface.co/datasets/AI-MO/NuminaMath-TIR

Table 4: Analysis of the impact of the mathematical code. The upper half presents the results of using and not using the mathematical code data. The lower half analyzes design of concatenating the reasoning steps and code snippets. "Basic + Reasoning-step-only" represents only adding the conditions, expressions, and results, while "Basic + Trans-code-only" represents only adding the translated code. "Basic + Separated Text&Code" represents seperating corresponding code and text. "Reasoning-Step&Code" represents the concatenated data combining both. "Basic + No-code-prompt" represents using a prompt that simply instruct Llama-3.1-70B-Instruct to rewrite texts to improve their quality.

| Data Composition | Base Model | MATH | GSM8K | SAT | OCW | MMLU-MATH |
|---|---|---|---|---|---|---|
| Basic | Llama-3-8B | 34.7 | 65.8 | 71.9 | 12.9 | 45.2 |
| Basic + Reasoning-Step&Code | Llama-3-8B | 38.4(+3.7) | 69.9(+4.1) | 84.4(+12.5) | 18.0(+5.1) | 46.5(+1.3) |
| Basic + Reasoning-step-only | DeepSeekCoder-1.3B | 16.7 | 22.7 | 40.6 | 4.8 | 25.9 |
| Basic + Trans-code-only | DeepSeekCoder-1.3B | 14.6 | 22.1 | 43.8 | 5.5 | 25.5 |
| Basic + No-code-prompt | DeepSeekCoder-1.3B | 15.7 | 21.3 | 37.5 | 4.8 | 24.4 |
| Basic + Separated Text&Code | DeepSeekCoder-1.3B | 17.0 | 22.0 | 46.9 | 4.8 | 25.3 |
| Basic + Reasoning-Step&Code | DeepSeekCoder-1.3B | **17.8** | **25.5** | **59.4** | **5.9** | **26.1** |

Table 5: Analysis of the effect of different components in MathCoder2-Pile. The base model is DeepSeekCoder-1.3B.

| Data Composition | MATH | GSM8K | SAT | OCW | MMLU-MATH |
|---|---|---|---|---|---|
| No Math Training | 4.8 | 4.3 | 18.8 | 2.6 | 24.8 |
| filtered-OpenWebMath (4.8B) | 9.0 | 11.4 | 34.4 | 3.7 | 25.4 |
| OpenWebMath (12.9B) | 9.4 | 11.2 | 31.3 | 2.6 | 24.4 |
| filtered-CC-En-math (6.4B) | 9.1 | 12.1 | 31.3 | 3.7 | 25.2 |
| CC-En-math (22.1B) | 8.4 | 13.0 | 25.0 | 2.9 | 25.0 |
| filtered-OpenWebMath + textbooks | 9.4 | 12.7 | 50.0 | 4.0 | 25.4 |
| filtered-OpenWebMath + synthetic data | 10.8 | 12.6 | 50.0 | 4.0 | 25.6 |
| filtered-OpenWebMath + code | 9.4 | 12.1 | 46.9 | 4.0 | 25.4 |
| MathCoder2-Pile | **17.8** | **25.5** | **59.4** | **5.9** | **26.1** |

improves performance across all five datasets. Note that the mathematical code only constitutes 14.1% of the 19.2B tokens in the MathCode-Pile dataset, yet the improvement in accuracy it brings about compared to the total improvement in accuracy ($\frac{acc_{\text{MathCode-Pile}} - acc_{\text{basic}}}{acc_{\text{MathCodePile}} - acc_{\text{orig}}}$) is 21.8%, 27.1%, 44.5%, 66.2%, and 35.1% on the five benchmark datasets, respectively, demonstrating the effectiveness of the mathematical code. Comparison across different training steps is shown in Appendix F.

We also analyze the design choice of concatenating the natural language reasoning step with the mathematical code for pretraining. This analysis is conducted by studying the results of adding only the natural language reasoning steps, and separately adding only the translated code. As shown in Tab. 4, Basic + Reasoning-step-only represents adding only the natural language reasoning steps; Basic + Trans-code-only represents adding only the translated code; Basic + Separated Text&Code represents seperating code and text; and Basic + Reasoning-Step&Code represents adding the concatenated data that combines both. The Basic + Reasoning-Step&Code configuration results in the best performance, demonstrating the importance of including both the natural language reasoning step and the translated mathematical code.

To rule out the possibility that the improvement comes from the higher quality of texts generated by Llama-3.1-70B-Instruct, we use a prompt that asks Llama-3.1-70B-Instruct to rewrite the given text. The details of this prompt are provided in Appendix E. We present the results of replacing the mathematical code with texts generated using this prompt in Tab. 4, labeled as "Basic + No-code-prompt". Our method of generating mathematical code accompanied with corresponding reasoning steps outperforms this baseline, demonstrating the effectiveness of our approach.

**Analysis of the impact of various parts of the basic data**. We perform experiments on a smaller model, DeepSeekCoder-1.3B, using different parts of the basic data. As demonstrated in

Table 6: Comparison between MathCode-Pile and other Mathematical Pretrain datasets.

| Pretrain Dataset | Base Model | MATH | GSM8K | SAT | OCW | MMLU-MATH |
|---|---|---|---|---|---|---|
| No Math Training | DeepSeekCoder-1.3B | 4.8 | 4.3 | 18.8 | 2.6 | 24.8 |
| OpenWebMath | DeepSeekCoder-1.3B | 9.4 | 11.2 | 31.3 | 2.6 | 24.4 |
| Proof-Pile-2 | DeepSeekCoder-1.3B | 9.2 | 11.2 | 50.0 | 4.4 | 25.8 |
| MathPile | DeepSeekCoder-1.3B | 5.3 | 3.4 | 21.9 | 2.2 | 24.9 |
| DeepSeekMath Corpus | DeepSeekLLM-1.3B | 13.6 | 23.8 | 56.3 | 4.8 | - |
| MathCoder2-Pile | DeepSeekCoder-1.3B | **17.8** | **25.5** | **59.4** | **5.9** | **26.1** |

Table 7: Comparison between finetuning the original Llama-3-8B, MathCoder2-Basic-Llama-3-8B, and MathCoder2-Llama-3-8B on NuminaMath-TIR. MathCoder2-Basic-Llama-3-8B is the model resulting from continued pretraining on the basic data.

| Base Model | MATH | GSM8K | OCW | Olympiad Bench | SVAMP |
|---|---|---|---|---|---|
| Llama-3-8B | 56.1 | 80.1 | 24.6 | 28.4 | 83.8 |
| MathCoder2-Basic-Llama-3-8B | 62.9 | 81.3 | 26.8 | 32.9 | 86.7 |
| MathCoder2-Llama-3-8B | **65.1** | **84.5** | **34.6** | **34.4** | **87.9** |

Tab. 5, filtered-OpenWebMath and filtered-CC-En-math significantly improve the performance of the model. In comparison, textbooks, synthetic data, and code are smaller in data size and play a less important role. As each of these parts of data is too small for individual pretraining, we combine them with OpenWebMath-filtered to show that they each bring a small yet noticeable improvement compared to using only OpenWebMath-filtered. Since we performed filtering on OpenWebMath and the initially filtered CC-En to remove irrelevant data, we also compare the performance before and after filtering. We observe that there is no obvious degradation in performance after removing irrelevant content, showing the effectiveness of the filtering.

**Comparison with other open-source mathematical pretraining corpora.** We compare MathCode-Pile with various other open-source mathematical pretraining corpora. We train each corpus for 3 epochs with a global batch size of 2 million tokens and a 4096 token context length, since we observe that the model's performance usually saturates around 3 epochs. As shown in Tab. 6, MathCode-Pile significantly outperforms OpenWebMath, Proof-Pile-2, and MathPile when trained on DeepSeekCoder-1.3B. The DeepSeekMath Corpus is not open-source, and its performance on DeepSeekLLM-1.3B is taken from Shao et al. (2024), which is trained for 150B tokens, more than our MathCode-Pile's training of approximately 60B tokens. The 1.3B model trained with MathCode-Pile outperforms the 1.3B model trained with DeepSeekMath Corpus.

**Analysis of the improvement on the potential of being finetuned for TIR reasoning.** To analyze the effect of the model-translated mathematical code on LLMs' potential to be finetuned for TIR reasoning, we finetune the original Llama-3-8B, MathCoder2-Basic-Llama-3-8B, and MathCoder2-Llama-3-8B on NuminaMath-TIR for three epochs, respectively. As shown in Tab. 7, the results of finetuning on MathCoder2-Basic-Llama-3-8B are higher than the results of finetuning on Llama-3-8B. Finetuning on MathCoder2-Llama-3-8B results in even higher performance than finetuning on MathCoder2-Basic-Llama-3-8B, showing that the addition of mathematical code effectively enhances the models' potential of being finetuned for TIR reasoning.

## 4 RELATED WORK

**Continued mathematical pretraining.** Several works (Shao et al., 2024; Azerbayev et al., 2024; Ying et al., 2024; Yang et al., 2024b) have explored the continued pretraining of LLMs on mathematical data, such as mathematical web content, synthetic data, and code. InternLM-Math (Ying et al., 2024) and Query of CC Fei et al. (2024) use BM25 for data retrieval, while other works such as DeepSeekMath (Shao et al., 2024) and Qwen2-Math (Yang et al., 2024b) employ fastText (Joulin et al., 2016) and other meta-information to retrieve texts from Common Crawl. Our approach follows these methods by using fastText for data filtering, and we introduce a second iteration of finer filtering to retain more relevant data. MathPile (Wang et al., 2023b) and phi (Gunasekar et al.,

2023) utilize real or synthesized textbooks, while Llemma (Azerbayev et al., 2024) and Qwen2-Math (Yang et al., 2024b) incorporate math-related code in their datasets. However, unlike our method of generating mathematical code with accompanied natural language reasoning, their code mostly has no natural language explanations or context. Our work builds on these prior efforts by collecting and expanding upon these sources of math-related text. Unlike works that only open-source their model weights, we take a more transparent approach by open-sourcing both our data processing and model training code, thereby ensuring reproducibility and facilitating future research in this field. Compared to Llemma (Azerbayev et al., 2024), which also open-source their data and training code, our method results in better performance on mathematical reasoning tasks.

**Synthetic data.** Numerous finetuning (Yu et al., 2024; Wang et al., 2023a; Lu et al., 2024a) and pre-training Gunasekar et al. (2023); Wang et al. (2023b); Yang et al. (2024b) studies have explored training on synthetic data generated using language models or predefined templates. MathGLM (Yang et al., 2023) and InternLM-Math (Ying et al., 2024) use templates to generate synthetic numerical operation data, while phi (Gunasekar et al., 2023) produces textbook-quality data with models. EntiGraph (Yang et al., 2024c) generates diverse text by drawing connections between sampled entities. Our work proposes a novel method for extracting mathematical reasoning steps and generating synthetic code snippets that captures the underlying reasoning processes.

**Post-training.** There are many methods for further improving the mathematical problem-solving abilities of LLMs. Supervised finetuning adjusts pretrained models using math problems and solutions in various formats, such as Chain-of-Thought (Yu et al., 2024; Yuan et al., 2023), Program-of-Thought (Yue et al., 2023), and Tool-Integrated Reasoning (Gou et al., 2024; Wang et al., 2023a; Liao et al., 2024). Reinforcement learning Lightman et al. (2023); Wang et al. (2024) and Direct Preference Optimization Rafailov et al. (2024); Xu et al. (2024); Lu et al. (2024b) utilize mathematical preference data to adjust the models' outputs. These methods are diverse and reveal the potential of pretrained models. Their performance is often influenced by the quality of the training data used in the pretraining stage. To explore the potential of finetuning our pretrained models for downstream tasks, we conduct supervised finetuning with existing open-source data.

## 5 LIMITATIONS AND FUTURE WORK

One limitation of our work is that our continued pretraining corpus focuses primarily on mathematics and does not intentionally include other STEM subjects. Additionally, our pretraining data consists entirely of English texts, without incorporating math-related content in other languages, like Chinese. Due to limitations in computational resources, we only trained models ranging from 1.3B to 8B parameters. Future work could address these limitations by expanding the dataset to include other subjects and languages and by training on larger language models. Also, this paper primarily focuses on continued mathematical pretraining, so we did not apply reinforcement learning methods like PPO and GRPO, or Direct Preference Optimization in our post-training phase, which can further improve performance on mathematical reasoning tasks. In the future, we could explore these methods on our finetuned models. Also, this work did not discuss theorem proving with formal languages such as Lean and Coq, which is worth investigating in future works.

## 6 CONCLUSION

In this paper, we present an effective open-source continued mathematical pretraining pipeline for enhancing mathematical reasoning of LLMs. Through the meticulous collection and filtering of diverse math-related texts, such as mathematical web content, synthetic data, code that uses mathematical packages, and math textbooks, we curate a basic dataset for continued mathematical pretraining. We then propose a novel method for extracting mathematical reasoning steps from the previously collected dataset and translating them to code snippets reflecting the underlying reasoning processes. By combining the basic data with the model-generated mathematical code accompanied with their corresponding reasoning steps, we produce a 19.2B-token mathematical pretraining corpus named MathCode-Pile, which significantly improves the performance of four different base models across five representative mathematical benchmarks. By open-sourcing the entire data processing pipeline and model training code, we actively promote transparency, reproducibility, and collaboration within the research community, facilitating future research in this area.

## 7 ACKNOWLEDGEMENT

This project is funded in part by National Key R&D Program of China Project 2022ZD0161100, by the Centre for Perceptual and Interactive Intelligence (CPII) Ltd under the Innovation and Technology Commission (ITC)'s InnoHK, by NSFC-RGC Project N_CUHK498/24. Hongsheng Li is a PI of CPII under the InnoHK.

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

## A  PROMPT FOR ANNOTATION OF MATH WEB DOCUMENTS

In this section, we present the prompt we used for annotation of documents in OpenWebMath and the initially filtered CC-En. The prompt, as shown in Fig. 3, asks the model to classify the document into one of seven types, which are types of documents that frequently appear in the datasets. We observe that this method helps the model to better identify and filter out irrelevant text than using a binary classification of whether the text is related to math.

## B  TRAINING DETAILS OF FASTTEXT CLASSIFIERS

We employ an open-source library[5] for training, configuring the vector dimension to 50, the learning rate to 0.5, the maximum length of word n-grams to 2, and the number of training epochs to 5. For the initial filtering of the Common Crawl corpus, we sample 3 million data points from the seed corpus of filtered-OpenWebMath as positive training examples and another 8 million web pages from Common Crawl as negative examples. For finer filtering, we use 2 million data points annotated as math-related by Mixtral-8x7B-Instruct as positive training samples and 1 million data points annotated as unrelated to math as negative training samples.

---

[5]https://fasttext.cc/

> **Prompt:** You will be provided with a block of text. I need you to classify the text into one of the following types:
>
> 1. The text describes a mathematical problem and its solution.
> 2. The text explains a mathematical concept or mathematical theory.
> 3. The text explains a scientific or engineering concept that requires mathematical knowledge.
> 4. The text describes a programming problem and its solution.
> 5. The text explains a concept or theory related to programming.
> 6. The text explains the usage of a programming language or software tool.
> 7. The text does not belong to any of the types above.
>
> Here's the text I've provided. Kindly analyze and classify it into type 1, 2, 3, 4, 5, 6 or 7. Put your choice behind "The type is:". Please do not generate any unrelated additional comments! The type number must match the type description. Here's one of the texts that needs to be classified: {TEXT}
> The type is:

Figure 3: The prompt for annotation of OpenWebMath and the initially filtered CC-En documents. {TEXT} is replaced with the content of the document.

> **Prompt:** You will be presented with a text related to math. I need you to carefully read through the text. If you find any incorrect statments, erroneous computation steps, spelling mistakes, grammatical errors, or formatting issues, adjust them so that the error is corrected. Rewrite the text to make it more accurate and easier to understand. You should only output an adjusted version of the given text. Also, do not change the original language. Please do not generate any unrelated additional comments! The text is as follows: {TEXT}
> You should output:

Figure 4: The prompt asking Llama-3.1-70B-Instruct to simply rewrite the text and improve its quality. {TEXT} is replaced with the content of the document.

## C    EXAMPLES OF THE MODEL-GENERATED MATHEMATICAL CODE

This session shows several examples of the translation from math-related texts to mathematical code acompanied with corresponding reasoning steps. As shown in Tab. 8, Tab. 9, Tab. 10 and Tab. 11, the model first extract the LaTex expression alone with its conditions and result from the original text, then generates an Python code snippet based on this information.

## D    EXAMPLES OF REMOVED IRRELEVANT TEXTS

In this section, we present several examples in the original OpenWebMath dataset that are irrelevant to mathematical reasoning and removed in the filtering process. As shown in Tab. 12, Tab. 13, and Tab. 14, the content of these documents are not related to math, but instead are about subjects such as politics, testing software, or web development. Removing these irrelevant texts have no obvious impact on the mathematical continued pretraining performance.

## E    PROMPT FOR SIMPLE REWRITING TO IMPROVE QUALITY FOR ABLATION STUDY

To rule out the possibility that the improvement results solely from the enhanced quality of the texts generated by Llama-3.1-70B-Instruct, we designed a prompt asking Llama-3.1-70B-Instruct to rewrite the text, checking for mistakes in content and format to enhance accuracy and clarity, as shown in Fig. 4.

---

**Original Text**:

Bayes and his Theorem

My earlier post on Bayesian probability seems to have generated quite a lot of readers, so this lunchtime I thought I'd add a little bit of background.

......

If we ask the question "what is the probability of exactly x successes from the possible n?", the answer is given by the binomial distribution:

$$P_n(x|n, p) = C(n, x)p^x(1 - p)^{n-x}$$

......

---

**Translation Output**:

Conditions Needed:

1. The number of trials (n) is a positive integer.
2. The probability of success (p) is a real number between 0 and 1.
3. The number of successes (x) is a non-negative integer less than or equal to n.

Computation Expression:

$P_n(x|n, p) = C(n, x)p^x(1 - p)^{n-x}$

Computation Result:

The probability of exactly x successes in n independent trials, each with a probability of success p.

Python Code Snippet:

```python
import math

def binomial_distribution(n, x, p):
    combinations = math.comb(n, x)
    probability = combinations * (p ** x) * ((1 - p) ** (n - x))
    return probability

# Example usage:
n = 10   # number of trials
x = 5    # number of successes
p = 0.5  # probability of success
result = binomial_distribution(n, x, p)
print(result)
```

---

Table 8: An example of translating a piece of text to mathematical code and a corresponding reasoning step.

## F    COMPARISON BETWEEN ADDING AND NOT ADDING MATHEMATICAL CODE

In this section, we present the comparison between adding and not adding mathematical code across different training steps. The experiments are conducted on Llama-3 8B. As shown in Fig. 5 and Fig. 6, adding the model-translated mathematical code improves accuracy across different training steps.

## G    REVISIONEFFECT OF MATHCODE-PILE ON GENERAL CODING AND OTHER ABILITIES

In order to explore MathCode-Pile's effect on general coding abilities, we test the MathCoder2 models on HumanEval and MBPP, two representative coding benchmarks, using the EvalPlus[6] frame-

---

[6]https://github.com/evalplus/evalplus

---

**Original Text**:
# Probability of getting 2 Aces, 2 Kings and 1 Queen in a five card poker hand (Part II)
So I reworked my formula in method 1 after getting help with my original question ......

$$\frac{\binom{4}{2}\binom{4}{2}\binom{4}{1}}{\binom{52}{5}} = \frac{3}{54145}$$

$$5.540678 * 10^{-5}$$

......

---

**Translation Output**:
Conditions Needed:
1. The total number of cards in a deck is 52.
2. We are drawing 5 cards from the deck.
3. We want to calculate the probability of getting 2 Aces, 2 Kings, and 1 Queen.
Computation Expression:

$$\frac{\binom{4}{2}\binom{4}{2}\binom{4}{1}}{\binom{52}{5}}$$

Computation Result:

$$5.540678 * 10^{-5}$$

Python Code Snippet:

```python
import math

def combination(n, k):
    return math.comb(n, k)

numerator = combination(4, 2) * combination(4, 2) * combination(4, 1)
denominator = combination(52, 5)

result = numerator / denominator
print(result)
```

---

Table 9: An example of translating a piece of text to mathematical code and a corresponding reasoning step.

work. HumanEval+ and MBPP+ are extended versions of HumanEval and MBPP with additional test samples, as introduced in Liu et al. (2023). The pass@1 accuracies are shown in Tab. 15. MathCode-Pile improves the performance of Llama3-8B, DeepSeekMath-7B, and Mistral-7B on these coding benchmarks. The performance of MathCoder2-CodeLlama-7B on these benchmarks is similar to that of CodeLlama-7B, which is understandable since CodeLlama is specifically trained for code generation. This demonstrates that MathCode-Pile can enhance general coding abilities beyond math-focused tasks for models not explicitly trained on code.

To demonstrate how MathCode-Pile affects LLMs' general abilities, we tested the MathCoder2 models on Hellaswag (Zellers et al., 2019), PIQA Bisk et al. (2019), and Winogrande Sakaguchi et al. (2019) using the lm-evaluation-harness[7]. As shown in the table below, training on MathCode-Pile slightly impacts the performance of general-purpose models, such as Llama3-8B and Mistral-7B, on these datasets, likely because MathCode-Pile consists entirely of math-related data. The effect on special-purpose models, such as DeepSeekMath-7B and CodeLlama-7B, is minimal.

---

[7]https://github.com/EleutherAI/lm-evaluation-harness

---

**Original Text**:
\# How do you use the product rule to differentiate $g(x) = (x^2 + 1)(x^2 - 2x)$?

......

and $h(x) = x^2 - 2x \Rightarrow h'(x) = 2x - 2$

......

---

**Translation Output**:
Conditions Needed:
1. $h(x)$ is a polynomial function
2. The derivative of $x^2$ is $2x$
3. The derivative of $-2x$ is $-2$
Computation Expression:
$\frac{d}{dx}(x^2 - 2x) = 2x - 2$
Computation Result:
$h'(x) = 2x - 2$
Python Code Snippet:

```python
import sympy as sp

x = sp.symbols('x')
h = x**2 - 2*x
h_prime = sp.diff(h, x)
print(h_prime)  # Output: 2*x - 2
```

---

Table 10: An example of translating a piece of text to mathematical code and a corresponding reasoning step.

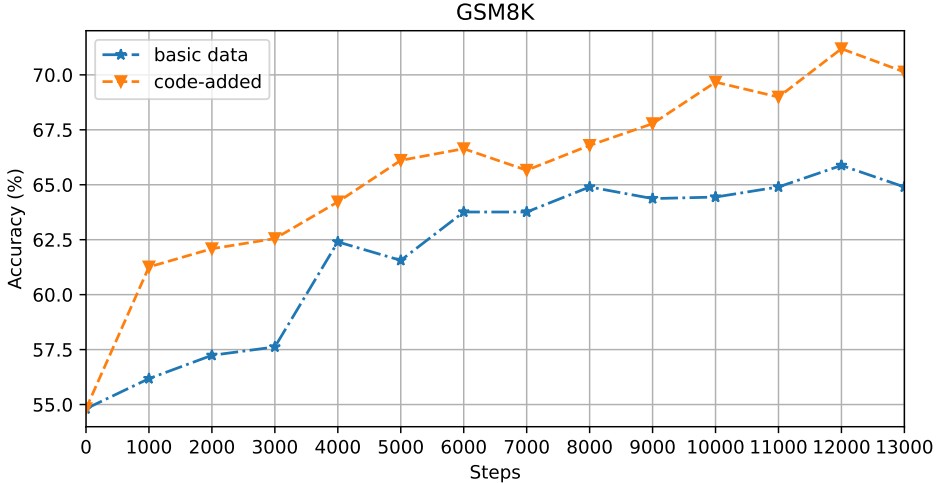

Figure 5: Comparison of the accuracy on GSM8K between adding and not adding mathematical code across different training steps.

## H  N-GRAMS OVERLAP BETWEEN TRAINING DATA AND TESTING DATA

We also apply n-gram testing to check for the overlapping ratio between MathCode-Pile and the test datasets, demonstrate the effectiveness of our decontamination. The overlaps at various n-grams are very low. The overlap becomes 0.00% when n-grams is 13.

**Original Text**:

\# How do you use the product rule to differentiate $g(x) = (x^2 + 1)(x^2 - 2x)$?

......

and $h(x) = x^2 - 2x \Rightarrow h'(x) = 2x - 2$

......

**Translation Output**:

Conditions Needed:

1. $h(x)$ is a polynomial function

2. The derivative of $x^2$ is $2x$

3. The derivative of $-2x$ is $-2$

Computation Expression:

$\frac{d}{dx}(x^2 - 2x) = 2x - 2$

Computation Result:

$h'(x) = 2x - 2$

Python Code Snippet:

```python
import sympy as sp

x = sp.symbols('x')
h = x**2 - 2*x
h_prime = sp.diff(h, x)
print(h_prime)  # Output: 2*x - 2
```

Table 11: An example of translating a piece of text to mathematical code and a corresponding reasoning step.

## Avoiding Weimar Russia

Matthew Yglesias writes:

Matthew Yglesias: Beyond Economics: Over at Brad DeLong's site you can see a fascinating discussion of America's Russia policy in the 1990s between DeLong, Martin Wolf, and Lawrence Summers. One remark I would make is that to an extraordinary extent, all three participants are willing to accept the premise that the only goal of US policy toward Russia in the 1990s was a good-faith effort to induce Russian prosperity, with such efforts being hampered by political constraints, the objective difficulty of the task, and pure policy errors...

Well, yes. Russia was once a superpower and may be one again. One would have thought that the history of 1914-1945 would teach ample lessons about the national security undesirability of trying to keep great powers–like Weimar Germany–poor and weak. One would have thought that the history of 1945-1990 would teach ample lessons about the national security desirability of trying to help great powers–like Japan and West Germany–become prosperous, democratic, and well-integrated into the world economy.

One top of the national-security strategic argument there is the economic argument: the fact that richer trading partners are better trading partners: they make more and more interesting stuff for us to buy.

......

Table 12: An example of removed text irrelevant to mathematical reasoning in OpenWebMath.

---

\# MicroEJ Test Suite Engine¶
\#\# Introduction¶
The MicroEJ Test Suite Engine is a generic tool made for validating any development project using automatic testing.
This section details advanced configuration for users who wish to integrate custom test suites in their build flow.
The MicroEJ Test Suite Engine allows the user to test any kind of projects within the configuration of a generic Ant file.
The MicroEJ Test Suite Engine is already pre-configured for running test suites on a MicroEJ Platform (either on Simulator or on Device).
\#\# Using the MicroEJ Test Suite Ant Tasks¶
Multiple Ant tasks are available in the testsuite-engine.jar provided in the Build Kit:
• testsuite allows the user to run a given test suite and to retrieve an XML report document in a JUnit format.
• javaTestsuite is a subtask of the testsuite task, used to run a specialized test suite for Java (will only run Java classes).
• htmlReport is a task which will generate an HTML report from a list of JUnit report files.
......

---

Table 13: An example of removed text irrelevant to mathematical reasoning in OpenWebMath.

---

By Kimserey Lam with
\# Conemu A Better Command Prompt For Windows
Jul 22nd, 2017 - written by Kimserey with .
When developing multiple Web api under multiple Visual Studio solutions, it can become very tedious to maintain, run and debug. Opening multiple instances of Visual Studio is very costly in term of memory and running all at once also clutter the screen which rapidly becomes irritating. With the advent of dotnet CLI tools, it has been clear that the next step would be to move out of the common "right click/build, F5" of Visual Studio and toward "dotnet run" on a command prompt. Last month I was looking for a Windows alternative of the bash terminal which can be found on Mac and I found ConEmu. ConEmu provides access to all typical shells via an enhanced UI. Today we will see how we can use ConEmu to ease our development process by leveraging only 2 of its features; the tasks and environment setup.
1. dotnet CLI 2. Setup environment 4. Apply to multiple services
......

---

Table 14: An example of removed text irrelevant to mathematical reasoning in OpenWebMath.

Table 15: Performance of the MathCoder2 models on general coding benchmarks: HumanEval, HumanEval+, MBPP and MBPP+, as well as general ability benchmarks: Hellaswag, PIQA and Winogrande.

| Model | Human-Eval | Human-Eval+ | MBPP | MBPP+ | Hella-swag | PIQA | Winog-rande |
|---|---|---|---|---|---|---|---|
| Llama-3-8B | 40.2 | 35.4 | 61.9 | 52.1 | **79.2** | **81.0** | **73.4** |
| **MathCoder2-Llama-3-8B** | **51.8** | **43.3** | **61.9** | **52.1** | 75.9 | 78.1 | 71.7 |
| DeepSeekMath-7B | 36.0 | 28.7 | 64.8 | 52.9 | 66.4 | **74.7** | **64.6** |
| **MathCoder2-DeepSeekMath-7B** | **36.6** | **32.3** | **66.7** | **54.8** | **66.9** | 74.0 | 63.1 |
| Mistral-7B | 29.3 | 23.8 | 51.3 | 40.5 | **81.1** | **82.0** | **73.9** |
| **MathCoder2-Mistral-7B** | **39.6** | **34.1** | **54.5** | **46.8** | 78.1 | 78.0 | 72.3 |
| Code-Llama-7B | 37.8 | **35.4** | **59.5** | 46.8 | **62.9** | **72.5** | **64.7** |
| **MathCoder2-Code-Llama-7B** | **38.4** | 32.3 | 58.5 | **47.4** | 62.8 | 72.3 | 63.7 |

Table 16: Overlap ratios for different n-grams.

| n-grams | 3 | 4 | 5 | 6 | 7 | 8 | 13 |
|---|---|---|---|---|---|---|---|
| **Overlap Ratio (%)** | 0.21 | 0.12 | 0.06 | 0.03 | 0.02 | 0.01 | 0.00 |

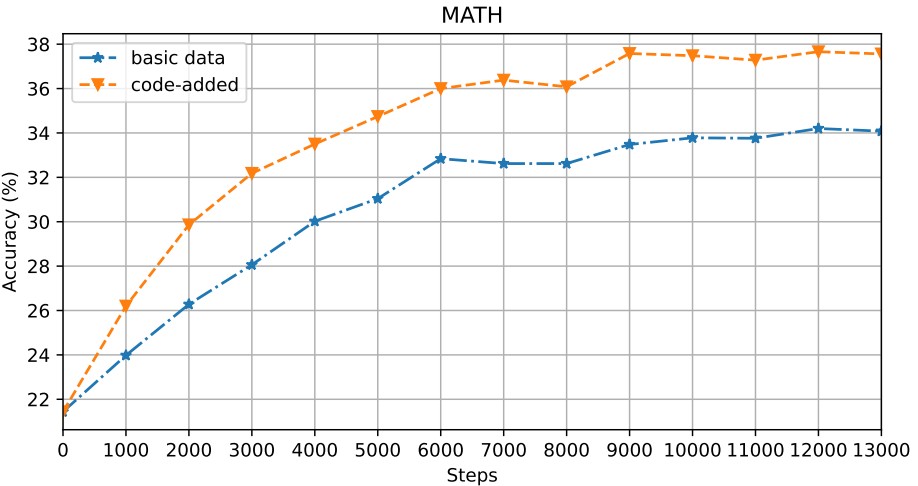

Figure 6: Comparison of the accuracy on MATH between adding and not adding mathematical code across different training steps.

