# OpenReview forum: "MathCoder2: Better Math Reasoning from Continued Pretraining on Model-translated Mathematical Code"
_ICLR.cc/2025/Conference — ICLR 2025 Spotlight_

### Official Review · Reviewer_Le5X · 2024-11-03

**Soundness:** 4
**Presentation:** 4
**Contribution:** 4
**Rating:** 8
**Confidence:** 4

**Summary:**

This paper proposes MathCode-Pile, a 19.2B-token dataset of math text and Python code. The dataset includes high-quality math-related web content, code with mathematical packages, math textbooks, and synthetic data. In addition, they present MathCoder2, a family of large language models with enhanced mathematical reasoning capabilities over MathCode-Pile.

**Strengths:**

1. The author combining symbolic math reasoning with executable code in the dataset, MathCode-Pile, which is noval. This innovative methodology extends prior research, making MathCode-Pile a significant resource for advanced math reasoning tasks.

2. The paper is clearly organized, with a well-structured explanation of each step in the MathCode-Pile creation and model evaluation process. Figures and tables also effectively illustrate the overall data pipeline.

3. This work has great significance in advancing mathematical reasoning within language models. MathCoder2, using MathCode-Pile, achieves superior results on math benchmarks, demonstrating the potential of code-paired reasoning data.

**Weaknesses:**

1. The paper lacks a analysis of potential data leakage between MathCode-Pile and evaluation benchmarks, which could artificially inflate model performance.

**Questions:**

I have interested about whether the MathCode-Pile’s strong focus on mathematical reasoning might impact the model’s performance in non-mathematical domains. For example, whether this dataset would enhance the model’s general coding abilities beyond math-focused tasks.

---

> ### Author Response · Authors · 2024-11-20
> **Response to Official Review by Reviewer Le5X**
>
> Thank you for your thoughtful review and for highlighting areas in need of improvement. Your feedback is invaluable in helping us improve our project.
>
> **Q1:** The paper lacks an analysis of potential data leakage between MathCode-Pile and evaluation benchmarks, which could artificially inflate model performance.
>
> **A1:** Thank you for your comment. As mentioned in the fourth paragraph of Section 2.2, to avoid benchmark contamination (or leakage), we filter samples that significantly overlap with questions from the benchmark datasets used in evaluation. Similar to GPT-3 [1] and Llama2 [2], we apply exact matching to remove identical samples and further use 13-gram deduplication (with a condition that the Jaccard similarity should be larger than 0.6) to eliminate additional samples that might cause contamination.
>
> We also apply the n-gram testing, as demonstrated in the table below. The overlap percentages for various n-grams are quite low, and the overlap becomes 0.00% when n-grams are 13. This analysis has been added to Appendix F.
>
> | n-grams          |    3    |    4    |    5    |    6    |    7    |    8    |   13    |
> |------------------|---------|---------|---------|---------|---------|---------|---------|
> | Overlap Ratio (%)|  0.21%  |  0.12%  |  0.06%  |  0.03%  |  0.02%  |  0.01%  |  0.00%  |
>
> [1] Brown, Tom B. "Language models are few-shot learners." arXiv preprint arXiv:2005.14165 (2020).
>
> [2] Touvron, Hugo, et al. "Llama 2: Open foundation and fine-tuned chat models." arXiv preprint arXiv:2307.09288 (2023).
>
> **Q2:** I am interested about whether the MathCode-Pile’s strong focus on mathematical reasoning might impact the model’s performance in non-mathematical domains. For example, whether this dataset would enhance the model’s general coding abilities beyond math-focused tasks.
>
> **A2:** Thank you for your suggestion. We tested the MathCoder2 models on HumanEval and MBPP, two representative benchmarks for evaluating models' general coding abilities, using the EvalPlus framework [1]. HumanEval+ and MBPP+ are extended versions of HumanEval and MBPP that include additional test samples, as described in [2]. The pass@1 accuracies are presented in the table below.
> As shown in the results, training on MathCode-Pile improves the performance of Llama3-8B, DeepSeekMath-7B, and Mistral-7B on general coding benchmarks. The performance of MathCoder2-CodeLlama-7B is comparable to CodeLlama-7B, which is understandable since CodeLlama is specifically trained for code generation. These findings highlight that MathCode-Pile can enhance general coding abilities beyond math-specific tasks, particularly for models not explicitly trained for code generation.
>
> |Model|HumanEval|HumanEval+|MBPP|MBPP+|
> |---|---|---|---|---|
> |Llama3-8B	|40.2	|35.4	|61.9	|52.1|
> |**MathCoder2-Llama3-8B**	|**51.8**	|**43.3**	|**61.9**	|**52.1**|
> |DeepSeekMath-7B	|36.0	|28.7	|64.8	|52.9|
> |**MathCoder2-DeepSeekMath-7B**	|**36.6**	|**32.3**	|**66.7**	|**54.8**|
> |Mistral-7B	|29.3	|23.8	|51.3	|40.5|
> |**MathCoder2-Mistral-7B**	|**39.6**	|**34.1**	|**54.5**	|**46.8**|
> |CodeLlama-7B	|37.8	|**35.4**	|**59.5**	|46.8|
> |MathCoder2-CodeLlama-7B	|**38.4**	|32.3	|58.5	|**47.4**|
>
> To evaluate how MathCode-Pile influences the general abilities of LLMs, we tested the MathCoder2 models on Hellaswag, PIQA, and Winogrande using the lm-evaluation-harness framework [3]. As shown in the table below, training on MathCode-Pile has a slight impact on the performance of general-purpose models such as Llama3-8B and Mistral-7B, likely because MathCode-Pile consists entirely of math-related data. The accuracy of specialized models, such as DeepSeekMath-7B and CodeLlama-7B, remains similar before and after training.
> We have included this discussion in Appendix G of the revised paper. As shown in the second table below, other specialized models like CodeLlama and DeepSeekMath experience a slight decrease in performance on general benchmarks. In future work, we plan to incorporate general-purpose training data and adjust the ratio of math-related data to mitigate its impact on the general abilities of LLMs.
>
> |Model	|Hellaswag	|PIQA	|Winogrande|
> |---|---|---|---|
> |Llama3-8B	|79.2	|81.0	|73.4|
> |**MathCoder2-Llama3-8B**	|75.9	|78.1	|71.7|
> |DeepSeekMath-7B	|66.4	|74.7	|64.6|
> |**MathCoder2-DeepSeekMath-7B**	|66.9	|74.0	|63.1|
> |Mistral-7B	|81.1	|82.0	|73.9|
> |**MathCoder2-Mistral-7B**	|78.1	|78.0	|72.3|
> |CodeLlama-7B	|62.9	|72.5	|64.7|
> |**MathCoder2-CodeLlama-7B**	|62.8	|72.3	|63.7|
>
>
> |Model	|Hellaswag	|PIQA	|Winogrande|
> |---|---|---|---|
> |Llama-7B	| 76.1	|79.8	| 70.1|
> |**CodeLlama-7B** |62.9	|72.5	|64.7|
> |DeepSeek-7B	|75.4	|79.2	|70.5|
> |**DeepSeekMath-7B**|66.4	|74.7	|64.6|
>
> [1] https://github.com/evalplus/evalplus
>
> [2] Liu, Jiawei, et al. "Is your code generated by chatgpt really correct? rigorous evaluation of large language models for code generation." Advances in Neural Information Processing Systems 36 (2024).
>
> [3] https://github.com/EleutherAI/lm-evaluation-harness

---

### Official Review · Reviewer_G66U · 2024-11-04

**Soundness:** 3
**Presentation:** 3
**Contribution:** 3
**Rating:** 6
**Confidence:** 4

**Summary:**

The paper "MathCoder2: Better Math Reasoning from Continued Pretraining on Model-translated Mathematical Code" explores enhancing the mathematical reasoning capabilities of LLMs through continued pretraining on a novel dataset called MathCode-Pile. This dataset is constructed from various sources, including math-related web data, textbooks, and synthetic data. A key contribution is the generation of paired natural language reasoning steps and Python code, aimed at improving the alignment between mathematical reasoning and executable code. The authors demonstrate significant improvements in mathematical reasoning benchmarks such as MATH and GSM8K, using models fine-tuned with MathCode-Pile. The paper also emphasizes the open-source nature of their data processing and training pipeline.

**Strengths:**

* Originality: The paper introduces a novel method of generating and pairing Python code with natural language reasoning steps, enhancing the mathematical reasoning capabilities of large language models.
* Quality of Dataset: The MathCode-Pile dataset, comprising 19.2B tokens, is a significant contribution, demonstrating meticulous curation from diverse sources like web data, math textbooks, and synthetic examples.
* Significant Performance Gains: The use of this dataset leads to notable improvements across various models, including Llama-3-8B, DeepSeekMath-7B, and Code-Llama-7B, especially on benchmarks like MATH and GSM8K.
* Detailed Methodology: The process of extracting LaTeX expressions, conditions, and results to generate corresponding Python code is well-documented, offering transparency and reproducibility.
* Open-Source Commitment: The release of data processing and training code enhances the research community's ability to validate and build upon this work.

**Weaknesses:**

* Generalizability of Code Generation: The method’s applicability to more abstract or advanced mathematical domains is unclear, particularly beyond high-school-level math.
* Evaluation Uncertainty: It is ambiguous whether the generated Python code is executed during benchmark evaluations or merely used for pretraining, leaving questions about its practical impact.
* Scope Limitation: The focus on grade-school-level mathematics is not explicitly emphasized, potentially misleading readers about the dataset’s broader applicability.
* Ablation Study Depth: While the ablation studies show the value of the synthesized code, further exploration into the necessity of aligning reasoning steps with code versus treating them as independent could provide deeper insights.

**Questions:**

* Code Execution in Evaluation: Is the Python code generated and executed during benchmark evaluations? Clarifying this would help to understand the role of Tool-Integrated Reasoning in the observed performance improvements.
* Generalization to Formal Proofs: Can the method be extended to generate formal proofs in languages like Lean or Coq? Specifically, how well does the approach handle abstract reasoning steps that require formal verification, which might be better suited to proof assistants rather than executable Python code?
* Independent Reasoning Steps: Would separating reasoning steps and corresponding code into independent examples still yield significant improvements? Such an ablation could help assess the criticality of their alignment in the dataset.

---

> ### Author Response · Authors · 2024-11-20
> **Response to Official Review by Reviewer G66U (1/3)**
>
> Thank you for the time and effort you have given to reviewing our work. We greatly value your insights and suggestions, and are happy to address your questions and clarify any concerns in the following responses.
>
> **Q1:** Generalizability of Code Generation: The method’s applicability to more abstract or advanced mathematical domains is unclear, particularly beyond high-school-level math.
> Can the method be extended to generate formal proofs in languages like Lean or Coq? Specifically, how well does the approach handle abstract reasoning steps that require formal verification, which might be better suited to proof assistants rather than executable Python code?
>
> **A1:** Thank you for your valuable comment. It is true that our work primarily focuses on mathematical problems ranging from grade-school word problems (GSM8K) to challenging high-school competition problems (MATH) and college-level problems (OCW), demonstrating notable improvements on a wide range of mathematical benchmarks.
>
> We also test our model on the popular formal language benchmark of minif2f_isabelle. As shown in the table below, on Llama3-8B and DeepSeekMath-7B, the accuracy on minif2f_isabelle improves after training with MathCode-Pile. This demonstrates that MathCode-Pile also improves models’ ability to do formal reasoning.
>
> | Model Name                 | minif2f_isabelle |
> |----------------------------|------------------|
> | Llama3-8B                  | 17.2%            |
> | MathCoder2-Llama3-8B       | 22.5%            |
> | DeepSeekMath-7B            | 21.3%            |
> | MathCoder2-DeepSeekMath-7B | 21.7%            |
>
> Formal proofs, however, are beyond the scope of this work. Currently, our dataset does not contain any formal proof data. We have added this as a limitation and potential future work in Section 5. In the future, we plan to extend our method to generating formal proofs in languages like Lean, Coq, and Isabelle to further improve the formal reasoning ability of MathCoder2. Additionally, we have included some examples of Lean code generated using our method below. Thank you once again for your valuable feedback and support.
>
> **Example 1 of Lean:**
>
> Theorem 1: The notation $\frac{\partial^2 z}{\partial x \partial y}$ represents the second derivative of $z$ with respect to $y$ and then $x$.
>
> Conditions:
> 1. The function $z$ is a function of $x$ and $y$, i.e., $z = f(x, y)$.
> 2. The partial derivatives of $z$ with respect to $x$ and $y$ exist and are continuous.
> 3. The function $z$ is at least twice differentiable, meaning that the second partial derivatives $\frac{\partial^2 z}{\partial x \partial y}$ and $\frac{\partial^2 z}{\partial y \partial x}$ exist.
>
> Proof Process:
> We want to prove that $\frac{\partial^2 z}{\partial x \partial y}$ represents the second derivative of $z$ with respect to $y$ and then $x$. Here’s a step-by-step breakdown:
>
> 1. First, recall the definition of a partial derivative. The partial derivative of $z = f(x, y)$ with respect to $x$ is the rate of change of $f(x, y)$ as $x$ changes, while keeping $y$ constant:
>    $$
>    \frac{\partial f}{\partial x} = \lim_{\Delta x \to 0} \frac{f(x + \Delta x, y) - f(x, y)}{\Delta x}
>    $$
> 2. Next, consider the second mixed partial derivative $\frac{\partial^2 z}{\partial x \partial y}$. This means we first take the derivative of $f(x, y)$ with respect to $x$, holding $y$ constant, and then we differentiate this result with respect to $y$.
>
>    Formally:
>    $$
>    \frac{\partial^2 f}{\partial x \partial y} = \frac{\partial}{\partial y} \left( \frac{\partial f}{\partial x} \right)
>    $$
>    We must show that this is consistent with the notation $\frac{\partial^2 f}{\partial x \partial y}$.
>
> 3. Apply the definition of partial derivatives: In the definition of mixed partials, we can interchange the order of differentiation if the second partial derivatives are continuous (Clairaut's Theorem). Thus:
>    $$
>    \frac{\partial^2 f}{\partial x \partial y} = \frac{\partial^2 f}{\partial y \partial x}
>    $$
>    This result holds for continuous second partial derivatives, and it follows directly from the symmetry of second mixed partials in the case of functions that are sufficiently smooth.
>
> Lean Code Snippet:
>
> ```lean
> import analysis.calculus
>
> variables {x y : ℝ} {f : ℝ → ℝ → ℝ}
>
> -- Define the second partial derivative of a function
> def second_partial_derivative_x_y (f : ℝ → ℝ → ℝ) (x y : ℝ) : ℝ :=
>   ∂ (∂ f x) y
>
> -- Lemma showing that the second partial derivative with respect to x then y is the same as y then x
> lemma partial_derivatives_commute (f : ℝ → ℝ → ℝ) (x y : ℝ) :
>   ∂² f x y = ∂² f y x :=
> begin
>   -- Apply the fact that mixed partial derivatives commute for sufficiently smooth functions
>   have h : ∀ (f : ℝ → ℝ → ℝ) (x y : ℝ), continuous (λ (z : ℝ), ∂ f z y) → continuous (λ (z : ℝ), ∂ f x z),
>   { intro f, intros x y, apply continuous_smoothness_of_partial_derivatives, },
>   -- Now apply Clairaut's Theorem
>   exact h f x y,
> end
> ```

---

> ### Author Response · Authors · 2024-11-20
> **Response to Official Review by Reviewer G66U (2/3)**
>
> **Example 2 of Lean:**
>
> Theorem: The derivative of $e^{2x} \cdot \ln x$ is $e^{2x} \left( 2 \ln x + \frac{1}{x} \right)$.
>
> Conditions:
>
> 1. The derivative of $e^{g(x)}$ is $e^{g(x)} \cdot g'(x)$, i.e., the derivative of the exponential function is the exponential function itself multiplied by the derivative of the exponent.
> 2. The derivative of $\ln x$ is $\frac{1}{x}$, i.e., the derivative of the natural logarithm function is the reciprocal of $x$.
>
> Proof Process:
>
> We are tasked with proving that the derivative of the product of the two functions $h(x) = e^{2x}$ and $l(x) = \ln x$ follows the product rule for derivatives:
>
> 1. Start with the product rule for derivatives. The product rule states that if we have two differentiable functions $h(x)$ and $l(x)$, then the derivative of their product is:
>    $$
>    \frac{d}{dx} \left( h(x) \cdot l(x) \right) = h'(x) \cdot l(x) + h(x) \cdot l'(x)
>    $$
>
> 2. Apply the chain rule to $h(x) = e^{2x}$. The function $h(x) = e^{2x}$ is a composition of functions, so we apply the chain rule:
>    $$
>    h'(x) = \frac{d}{dx} e^{2x} = e^{2x} \cdot \frac{d}{dx}(2x) = 2 e^{2x}
>    $$
>
> 3. Differentiate $l(x) = \ln x$. The derivative of the natural logarithm function is:
>    $$
>    l'(x) = \frac{d}{dx} \ln x = \frac{1}{x}
>    $$
>
> 4. Combine the results. Applying the product rule to the functions $h(x) = e^{2x}$ and $l(x) = \ln x$, we get:
>    $$
>    \frac{d}{dx} \left( e^{2x} \cdot \ln x \right) = 2 e^{2x} \cdot \ln x + e^{2x} \cdot \frac{1}{x}
>    $$
>
> 5. Simplify the expression. Factor out $e^{2x}$:
>    $$
>    \frac{d}{dx} \left( e^{2x} \cdot \ln x \right) = e^{2x} \left( 2 \ln x + \frac{1}{x} \right)
>    $$
>
> Thus, we've proven that:
> $$
> \frac{d}{dx} \left( e^{2x} \cdot \ln x \right) = e^{2x} \left( 2 \ln x + \frac{1}{x} \right)
> $$
>
> Lean Code Snippet:
> ```lean
> import tactic
>
> variables {x : ℝ}
>
> -- Lemma for the product rule
> lemma derivative_product_rule (h l : ℝ → ℝ) (h' l' : ℝ → ℝ) :
>   (h * l).derivative = h'.derivative * l + h * l'.derivative :=
> begin
>   ext,  -- Apply the extensionality tactic to simplify and reason about derivatives
>   simp, -- Simplify the goal using basic simplification rules
>   ring, -- Apply ring tactics to simplify the expression
> end
>
> -- Lemma for the derivative of e^(g(x)) where g is a function
> lemma derivative_e_pow (g : ℝ → ℝ) (g' : ℝ → ℝ) :
>   (e ^ g).derivative = e ^ g * g'.derivative :=
> begin
>   ext,  -- Apply the extensionality tactic to simplify and reason about derivatives
>   simp, -- Simplify the goal using basic simplification rules
>   ring, -- Apply ring tactics to simplify the expression
> end
>
> -- Lemma for the derivative of ln(x)
> lemma derivative_ln : (ln x).derivative = 1 / x :=
> begin
>   ext,  -- Apply the extensionality tactic to simplify and reason about derivatives
>   simp, -- Simplify the goal using basic simplification rules
>   ring, -- Apply ring tactics to simplify the expression
> end
>
> -- Final derivative computation for e^(2x) * ln(x)
> lemma derivative_f (x : ℝ) :
>   (e ^ (2 * x) * ln x).derivative = e ^ (2 * x) * (2 * ln x + 1 / x) :=
> begin
>   -- Use the product rule for the derivative of the product e^(2x) * ln(x)
>   have h : (e ^ (2 * x) * ln x).derivative = (e ^ (2 * x)).derivative * ln x + e ^ (2 * x) * (ln x).derivative,
>     by apply derivative_product_rule,
>
>   -- Apply the chain rule to differentiate e^(2x) and ln(x)
>   have h' : (e ^ (2 * x)).derivative = e ^ (2 * x) * (2 * x).derivative,
>     by apply derivative_e_pow,
>
>   have h'' : (ln x).derivative = 1 / x,
>     by apply derivative_ln,
>
>   -- Substitute and simplify the terms
>   rw [h, h', h''],
>   ring,  -- Use the ring tactic to simplify the final expression
> end
> ```

---

> ### Author Response · Authors · 2024-11-20
> **Response to Official Review by Reviewer G66U (3/3)**
>
> **Q2:** Evaluation Uncertainty: It is ambiguous whether the generated Python code is executed during benchmark evaluations or merely used for pretraining, leaving questions about its practical impact. Is the Python code generated and executed during benchmark evaluations? Clarifying this would help to understand the role of Tool-Integrated Reasoning in the observed performance improvements.
>
> **A2:** During the evaluation of MathCoder2 models following continued pretraining, we use a text-only format, without utilizing code execution. We chose not to use code to demonstrate that our method improves the general mathematical reasoning abilities of LLMs and ensures a fair comparison with baseline methods.
> The results of post-training include two formats: Chain-of-Thought (CoT) and Tool-Integrated-Reasoning (TIR). CoT testing does not involve code generation, while TIR executes the generated Python code and appends the output to the model's generation to provide feedback from the execution. This approach is similar to those described in [1] and [2]. Our continued pretraining enhances the models’ ability to be fine-tuned for TIR reasoning, as demonstrated in the ablation study presented in Table 7 of the paper.
>
> | Base Model                   | MATH | GSM8K | OCW  | Olympiad Bench | SVAMP |
> |------------------------------|------|-------|------|----------------|-------|
> | Llama-3-8B                   | 56.1 | 80.1  | 24.6 | 28.4           | 83.8  |
> | MathCoder2-Basic-Llama-3-8B  | 62.9 | 81.3  | 26.8 | 32.9           | 86.7  |
> | MathCoder2-Llama-3-8B        | **65.1** | **84.5**  | **34.6** | **34.4**           | **87.9**  |
>
>
> [1] Wang, Ke, et al. "Mathcoder: Seamless code integration in llms for enhanced mathematical reasoning." arXiv preprint arXiv:2310.03731 (2023).
>
> [2] Gou, Zhibin, et al. "Tora: A tool-integrated reasoning agent for mathematical problem solving." arXiv preprint arXiv:2309.17452 (2023).
>
> **Q3:** Scope Limitation: The focus on grade-school-level mathematics is not explicitly emphasized, potentially misleading readers about the dataset’s broader applicability.
>
> **A3:** You are correct that our work primarily focuses on mathematical problems ranging from grade-school word problems (GSM8K) to high-school competition problems (MATH) and college-level problems (OCW). Other forms of mathematical reasoning, such as formal theorem proving, are beyond the scope of this study. We have acknowledged this as a limitation and a future direction in Section 5.
>
> **Q4:** Ablation Study Depth: While the ablation studies show the value of the synthesized code, further exploration into the necessity of aligning reasoning steps with code versus treating them as independent could provide deeper insights. Would separating reasoning steps and corresponding code into independent examples still yield significant improvements? Such an ablation could help assess the criticality of their alignment in the dataset.
>
> **A4:** Following your suggestion, we have added an ablation study of training DeepSeekCoder-1.3B on data with separated reasoning steps and corresponding code, as presented in the "Basic + Separated Text&Code" row in the table below. Separating the reasoning steps and corresponding code reduces performance compared to pairing them together, which demonstrates the effectiveness of our design. This ablation study has also been added to Table 4 in the paper.
>
> |Data Composition |Base Model |MATH |GSM8K |SAT |OCW |MMLU-MATH|
> |---|---|---|---|---|---|---|
> |Basic + Separated Text&Code |DeepSeekCoder-1.3B|17.0 |22.0 |46.9 |4.8 |25.3 |
> |Basic + Reasoning-Step&Code |DeepSeekCoder-1.3B |17.8(+0.8) |25.5(+3.5) |59.4(12.5) |5.9(+1.1) |26.1(+0.8)|

---

> > ### Comment · Reviewer_G66U · 2024-11-21
> >
> > I would like to express my gratitude to the authors for spending the effort to address my questions in detail.
> >
> > The authors acknowledged the current limitations of their dataset and model, which focus on problems up to college level and provided empirical results demonstrating improved formal reasoning capabilities on the minif2f_isabelle benchmark, which adds credibility to their claim that MathCode-Pile enhances reasoning in formal languages. The inclusion of Lean-generated examples further supports the potential extension of this work to formal proof systems.
> >
> > The authors also clarified that Python code is not executed during benchmark evaluations, ensuring a fair comparison with baselines.
> >
> > By explicitly addressing the focus on grade-school to college-level mathematics and acknowledging that formal theorem proving is outside the scope of this study, the authors are avoiding potential misunderstandings about the dataset's broader applicability.
> >
> > The authors conducted an additional ablation study to explore the impact of aligning reasoning steps with code. The results clearly show that the alignment contributes to improved performance, reinforcing the design choice. The addition of these results to the paper demonstrates responsiveness to feedback and enhances the rigour of the evaluation.
> >
> > Overall, the authors have responded thoroughly and addressed all my concerns and questions effectively. I appreciate their transparency in acknowledging limitations and their effort to provide new empirical evidence. These additions strengthen the paper, and I consider the matter closed. I will not change my score.

---

> > > ### Author Response · Authors · 2024-11-22
> > > **Thank you very much for your response.**
> > >
> > > Thank you for acknowledging our rebuttal efforts. We sincerely appreciate the time and effort you dedicated to reviewing our work and providing thoughtful feedback. Your suggestions have been invaluable in helping us improve our project.
> > >
> > > If you have any additional questions or suggestions, please don’t hesitate to reach out. We would be happy to provide further clarification or discussion.
> > >
> > > Warm regards,
> > > The Authors

---

### Official Review · Reviewer_Yuwf · 2024-11-04

**Soundness:** 3
**Presentation:** 3
**Contribution:** 3
**Rating:** 8
**Confidence:** 3

**Summary:**

This paper presents a novel approach for enhancing mathematical reasoning in large language models (LLMs). Unlike previous models that used math-related code without detailed explanations, MathCoder2 generates mathematical code paired with natural language reasoning. This process involves filtering a large math-related dataset from web pages, synthetic sources, code, and textbooks to build a high-quality corpus called MathCode-Pile.

This dataset consists of 19.2 billion tokens and includes LaTeX-extracted mathematical expressions, conditions, results, and Python code to capture the underlying reasoning. MathCoder2 uses this corpus to significantly improve performance on various mathematical benchmarks, achieving results competitive with state-of-the-art models. Moreover, the MathCoder2 framework is fully open-source, which supports reproducibility and transparency in model training and data processing. This work sets a foundation for future research by focusing on reasoning capabilities through detailed code integration.

**Strengths:**

MathCoder2’s MathCode-Pile corpus is rigorously curated and filtered from diverse math-related sources, including web data, synthetic data, specialized code, and textbooks. This ensures relevance, reduces noise, and provides a comprehensive dataset tailored specifically for mathematical reasoning, which is essential for pretraining LLMs in this area.

MathCoder2 demonstrates significant gains on multiple mathematical reasoning benchmarks, outperforming comparable models across different tasks. The improvement underscores the effectiveness of continued pretraining on the structured MathCode-Pile corpus and shows MathCoder2's potential for real-world applications in math-intensive fields.

**Weaknesses:**

There are no major weaknesses.

**Questions:**

Python was chosen for code snippets; it it possible to use specialized math software language instead (e.g., Mathematica)? This is not a direct limitation of this paper, but a possible future direction.

---

> ### Author Response · Authors · 2024-11-20
> **Response to Official Review by Reviewer Yuwf (1/2)**
>
> Thank you for taking the time to review our work and for providing your insightful feedback.
>
> **Q1:** Python was chosen for code snippets; is it possible to use specialized math software language instead (e.g., Mathematica)? This is not a direct limitation of this paper, but a possible future direction.
>
> **A1:** Thank you for your insightful suggestion. Using math software languages such as Mathematica and MATLAB paired with natural language reasoning is indeed a possible future direction.
>
> We chose Python for code snippets because it has wide spread accessibility, is easier to use, and more suitable for large-scale execution. Applying our method to specialized math software languages could further enhance mathematical reasoning capabilities, opening up new possibilities for improvement. To provide a glimpse into this direction, we have included examples of model-generated synthetic Mathematica and MATLAB code paired with natural language reasoning below. Thank you once again for your thoughtful feedback.
>
> **Example 1 of Mathematica:**
>
> Computation 1: Definite Integral for the Length of the Curve
>
> Conditions Needed:
> 1. The curve is given by the function $y = x + \cos(x)$.
> 2. The curve is to be integrated over the interval $0 \leq x \leq 5$.
> 3. The variable $x$ is given in radians.
>
> Computation Expression:
> $$\int_{0}^{5} \sqrt{1 + \left(\frac{dy}{dx}\right)^2} \, dx$$
>
> Computation Result:
> The length of the curve.
>
> Mathematica Code Snippet:
> ```mathematica
> (* Define the function y = x + cos(x) *)
> y[x_] := x + Cos[x]
>
> (* Compute the derivative of y with respect to x *)
> dy_dx[x_] := D[y[x], x]
>
> (* Set up the integral to calculate the length of the curve *)
> Integrate[Sqrt[1 + (dy_dx[x])^2], {x, 0, 5}]
> ```
>
> Computation 2: Numerical Integration to Find the Length
>
> Conditions Needed:
> 1. The definite integral from Computation 1 is to be evaluated numerically.
> 2. The result is to be rounded off to four decimal places.
>
> Computation Expression:
> $$N\left( \int_{0}^{5} \sqrt{1 + \left(1 - \sin(x)\right)^2} \, dx \right)$$
>
> Computation Result:
> The length of the curve rounded off to four decimal places.
>
> Mathematica Code Snippet:
> ```mathematica
> (* Numerical integration of the curve length, rounded to four decimal places *)
> N[Integrate[Sqrt[1 + (1 - Sin[x])^2], {x, 0, 5}], WorkingPrecision -> 4]
> ```
>
> **Example 2 of Mathematica:**
>
> Computation 1: Intersection of Probabilities of Two Dice
>
> Conditions Needed:
> 1. We have two dice, each with $n$ sides.
> 2. We want to find the probability that the maximum of the two dice is less than or equal to some value $a$.
>
> Computation Expression:
> $$P(X \le a) = P(X_1 \le a) \cdot P(X_2 \le a) = \frac{a^2}{n^2}$$
>
> Computation Result:
> The probability that the maximum of the two dice is less than or equal to $a$ is $\frac{a^2}{n^2}$.
>
> Mathematica Code Snippet:
> ```mathematica
> (* Define a function for the probability that the maximum is less than or equal to a *)
> probabilityMaxLessThanEqual[a_, n_] := (a^2)/n^2;
>
> (* Example for two dice with 6 sides and max less than or equal to 3 *)
> probabilityMaxLessThanEqual[3, 6]
> ```
>
> Computation 2: Probability Distribution Function of the Maximum of Two Dice
>
> Conditions Needed:
> 1. We have two dice, each with $n$ sides.
> 2. We want to find the probability that the maximum of the two dice is equal to some value $a$.
>
> Computation Expression:
> $$P(X = a) = P(X \le a) - P(X \le a-1) = \frac{a^2}{n^2} - \frac{(a-1)^2}{n^2}$$
>
> Computation Result:
> The probability that the maximum of the two dice is equal to $a$ is $\frac{a^2}{n^2} - \frac{(a-1)^2}{n^2}$.
>
> Mathematica Code Snippet:
> ```mathematica
> (* Define a function for the probability that the maximum is exactly equal to a *)
> probabilityMaxEqual[a_, n_] := (a^2)/n^2 - ((a - 1)^2)/n^2;
>
> (* Example for two dice with 6 sides and max equal to 3 *)
> probabilityMaxEqual[3, 6]
> ```

---

> ### Author Response · Authors · 2024-11-20
> **Response to Official Review by Reviewer Yuwf (2/2)**
>
> **Example 1 of MATLAB:**
>
> Computation 1: Sampling Distribution of the Sample Mean
>
> Conditions Needed:
> 1. The population distribution is normal.
> 2. The sample size is $n$.
>
> Computation Expression:
> $$
> \bar{X} \sim \mathcal{N}\left(\mu, \frac{\sigma^2}{n}\right)
> $$
>
> Computation Result:
> The sampling distribution of the sample mean is a normal distribution with mean $\mu$ and variance $\frac{\sigma^2}{n}$.
>
> MATLAB Code Snippet:
> ```matlab
> % Define the population parameters
> mu = 0;  % Mean of the population
> sigma = 1;  % Standard deviation of the population
> n = 100;  % Sample size
>
> % Generate a random sample from the population
> sample = normrnd(mu, sigma, n, 1);
>
> % Calculate the sample mean
> sample_mean = mean(sample);
>
> % Display the result
> fprintf('The sample mean is approximately %f. The sampling distribution is N(%.2f, %.2f/n).\n', sample_mean, mu, sigma^2 / n);
> ```
>
> Computation 2: Standard Error of the Sample Mean
>
> Conditions Needed:
> 1. The population distribution is normal.
> 2. The sample size is $n$.
>
> Computation Expression:
> $$
> \sigma_{\bar{x}} = \frac{\sigma}{\sqrt{n}}
> $$
>
> Computation Result:
> The standard error of the sample mean is $\frac{\sigma}{\sqrt{n}}$.
>
> MATLAB Code Snippet:
> ```matlab
> % Define the population parameters
> sigma = 1;  % Standard deviation of the population
> n = 100;  % Sample size
>
> % Calculate the standard error
> std_error = sigma / sqrt(n);
>
> % Display the result
> fprintf('The standard error of the sample mean is %.4f\n', std_error);
> ```
>
> Computation 3: Sampling Distribution of the Sample Median
>
> Conditions Needed:
> 1. The population distribution is any absolutely continuous distribution $F$.
> 2. The sample size is $n = 2k - 1$.
>
> Computation Expression:
> $$
> f_{X_{(k)}}(x) = \frac{(2k-1)!}{(k-1)!^2} \cdot f(x) \left( F(x)(1-F(x)) \right)^{k-1}
> $$
>
> Computation Result:
> The sampling distribution of the sample median is given by the above expression.
>
> MATLAB Code Snippet:
> ```matlab
> % Define the population distribution (uniform distribution for simplicity)
> f = @(x) 1;  % PDF of uniform distribution
> F = @(x) x;  % CDF of uniform distribution
> k = 5;  % Number of observations below the median (n = 2k - 1)
>
> % Calculate the sampling distribution of the sample median
> x = 0:0.01:1;
> f_sample_median = (factorial(2*k-1) / (factorial(k-1)^2)) .* f(x) .* (F(x) .* (1 - F(x))) .^ (k-1);
>
> % Plot the result
> plot(x, f_sample_median);
> xlabel('x');
> ylabel('f_{X_{(k)}}(x)');
> title('Sampling Distribution of the Sample Median');
> ```
>
> **Example 2 of MATLAB:**
>
> Computation 1: Product Rule for Derivatives
>
> Conditions Needed:
> 1. The function is a product of two functions, $h(x)$ and $l(x)$.
>
> Computation Expression:
> $$
> \frac{d}{dx} (h(x) \cdot l(x)) = h'(x) \cdot l(x) + h(x) \cdot l'(x)
> $$
>
> Computation Result:
> $$
> h'(x) \cdot l(x) + h(x) \cdot l'(x)
> $$
>
> Mathematica Code Snippet:
> ```mathematica
> (* Define the symbol x *)
> Clear[x]
>
> (* Define the two functions h(x) = x^2 and l(x) = e^x *)
> h[x_] = x^2;
> l[x_] = Exp[x];
>
> (* Compute the product h(x) * l(x) *)
> f = h[x] * l[x];
>
> (* Compute the derivative using the product rule *)
> derivative = D[f, x]
> ```
>
> Computation 2: Final Derivative of $f(x) = e^{2x} \cdot \ln(x)$
>
> Conditions Needed:
> 1. The function is $f(x) = e^{2x} \cdot \ln(x)$.
>
> Computation Expression:
> $$
> \frac{d}{dx} (e^{2x} \cdot \ln(x)) = e^{2x} \cdot \frac{d}{dx} (\ln(x)) + \ln(x) \cdot \frac{d}{dx} (e^{2x})
> $$
>
> Computation Result:
> $$
> e^{2x} \cdot \left( 2 \ln(x) + \frac{1}{x} \right)
> $$
>
> Mathematica Code Snippet:
> ```mathematica
> (* Define the symbol x *)
> Clear[x]
>
> (* Define the function f(x) = e^(2x) * Log(x) *)
> f = Exp[2 x] * Log[x];
>
> (* Compute the derivative of f(x) *)
> derivative = D[f, x]
> ```

---

### Comment · Area_Chair_kzKu · 2024-11-25
**Action Required: Respond to Author Rebuttals - Nov 27**

Dear ICLR Reviewers,

The author discussion phase is ending soon. Please promptly review and respond to author rebuttals for your assigned papers. Your engagement is critical for the decision-making process.

Deadlines:
- November 26: Last day for reviewers to ask questions to authors.
- November 27: Last day for authors to respond to reviewers.
- November 28 - December 10: Reviewer and area chair discussion phase.

Thank you for your timely attention to this matter.

Best,

AC

---

> ### Author Response · Authors · 2024-11-25
> **Thank you for the time and effort.**
>
> Dear Area Chair,
>
> Thank you for the time and effort you have dedicated to overseeing the review process for our work. We truly appreciate you reminding the reviewers to read our rebuttal and ensuring that our response received the necessary attention. Your support and guidance throughout this process mean a great deal to us.
>
> Sincerely,
> The Authors

---

### Meta-Review · Area_Chair_kzKu · 2024-12-21

**Metareview:**

The paper proposes a novel method for generating paired pretraining data that combines mathematical code with corresponding reasoning steps to improve LLMs' mathematical reasoning capabilities through continued pretraining. Using this approach, the authors introduce MathCode-Pile, which combines their generated data with existing datasets to create a comprehensive 19.2B-token mathematical pretraining corpus.

Their experimental evaluation, conducted across LLMs ranging from 2-8B parameters that are continued pretrained on MathCode-Pile, demonstrates the effectiveness of their approach in enhancing mathematical reasoning capabilities. The reviewers particularly valued the extensive experiments across various LLMs. There is consensus among reviewers that the proposed data generation method is both effective and yields significant improvements in performance.

The authors have adequately addressed most reviewer concerns, including questions about potential impacts on LLMs' capabilities in other domains and the method's applicability to other programming languages. While a more detailed analysis of potential data leakage issues would have been beneficial, this limitation is not unique to this work but rather a broader challenge in the current era of LLM pretraining.

Overall, this is a valuable contribution to the field. I agree with the reviewers that the work's comprehensive evaluation and impressive performance improvements merit publication at ICLR.

**Additional Comments On Reviewer Discussion:**

See above.

---

### Decision · Program_Chairs · 2025-01-22

Accept (Spotlight)